# Uncalibrated Reasoning: GRPO Induces Overconfidence for Stochastic Outcomes

## Abstract

Reinforcement learning (RL) has proven remarkably effective at improving the accuracy of language models in verifiable and deterministic domains like mathematics and coding. However, it is unclear if current RL methods are similarly effective at optimizing language models to reason about the probability of uncertain events from stochastic data, a valuable capability for decision-making and scientific discovery. Here, we demonstrate that Group Relative Policy Optimization (GRPO) induces highly overconfident probability predictions across three proper scoring rule rewards, while Proximal Policy Optimization (PPO) and REINFORCE Leave-One-Out (RLOO) yield well-calibrated models. We show that removing group standard normalization in GRPO fixes its miscalibration and provide a theoretical explanation for why GRPO's biased advantage estimate causes overconfidence. Our results demonstrate the negative impact of GRPO's standard normalization on probabilistic prediction and highlight an important design consideration for RL algorithms: while unbiased advantage estimates provide a consistent optimization signal across tasks, biased advantage estimates must be aligned with the structure of the target objective to be effective.

## 1 Introduction

Reinforcement learning (RL) has achieved remarkable success at improving the accuracy of language models in verifiable domains like mathematics and coding (OpenAI, 2024; Shao et al., 2024; Kimi Team, 2025). In particular, recent success has been achieved by optimizing language models to generate chain-of-thought text before responding to a prompt, often called "reasoning", with supervision from a verifier. Current research has focused primarily on domains where answers are deterministically correct or incorrect.

An important next step for the reasoning RL paradigm is probabilistic reasoning: optimizing models to reason about the probability of uncertain events. Calibrated probability estimates are important for optimal decision making (Zhao et al., 2021): for example, it is helpful for a model assisting in medical decision support to distinguish between diagnoses with a 60% and 99% chance of being correct. Moreover, moving beyond deterministic verifiers to learn from stochastic supervision could greatly expand the available training signal for optimizing reasoning models. For example, scientific experiments, which are subject to random variation, form a natural source of probabilistic supervision that extends far beyond current written knowledge.

In this paper, we study whether three RL algorithms widely used to optimize reasoning models, namely GRPO (Shao et al., 2024), PPO (Schulman et al., 2017), and RLOO (Kool et al., 2019; Ahmadian et al., 2024), are also effective at optimizing models to predict the probability of uncertain outcomes. Across applications like medical question answering and real-world scientific experiments, we demonstrate that models optimized with GRPO using a log-likelihood reward make highly overconfident probability predictions, while models optimized with PPO, RLOO, and GRPO without group standard normalization are well-calibrated. Interestingly, we find that models optimized with each algorithm achieve similar accuracy based on their maximum probability prediction, despite differences in calibration and rewards. To test the generality of this effect, we construct a minimal synthetic setting and show that GRPO's overconfidence persists across alternative proper scoring rules, non-transformer architectures, and both token-based and continuous uncertainty parameterizations. Finally, we provide a theoretical analysis showing how standard normalization in GRPO's advantage

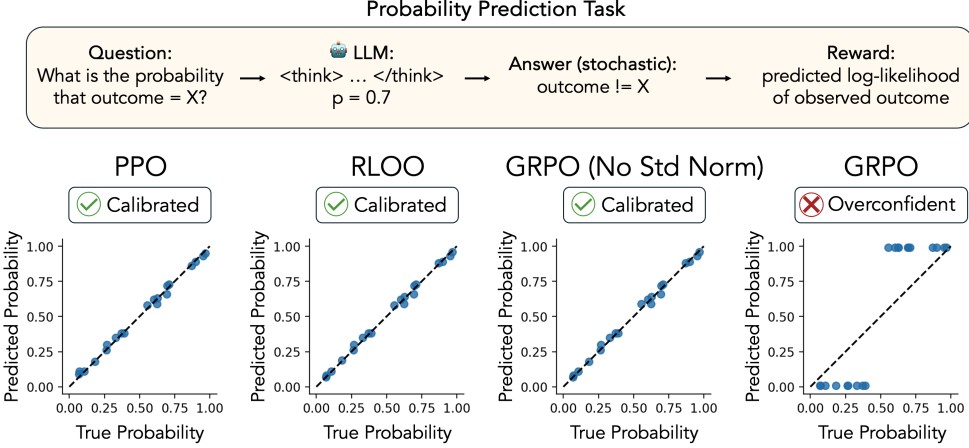

Figure 1: Group standard normalization in GRPO induces overconfident predictions of stochastic outcome probabilities. **Top:** Probability prediction task. **Bottom:** Synthetic data experiment results. Models trained with PPO, RLOO, and GRPO with no standard normalization are well calibrated, while models trained with GRPO are extremely overconfident.

estimate drives systematic overconfidence. Taken together, our results demonstrate that group standard normalization introduces a harmful bias when optimizing probabilistic predictions. More broadly, the divergence between GRPO's success in accuracy tasks and its failures in probability estimation reveals an important principle: unbiased advantage estimates provide consistent optimization signal across tasks, whereas biased estimators must be aligned with the structure of the target objective to be effective.

## 2 PRELIMINARIES

**RL with Language Models** Reinforcement learning methods cast autoregressive language models as stochastic policies $\pi_\theta$ that specify actions (selecting new tokens) based on the current state (the prompt and prior generated tokens). We consider a setting with outcome supervision, where the goal is to maximize the expected reward received from a verifier that scores the correctness of a response given the ground-truth answer. While current work focuses primarily on settings with deterministic answers, we consider answers that may be stochastic conditional on the prompt.

**Value and Advantage Functions** The state value function $V^\pi(s)$ is defined as the expected reward from following policy $\pi$ from state $s$, and the state-action value function $Q^\pi(s, a)$ is defined as the expected reward of following policy $\pi$ from state $s$ when the next action is set to be $a$. The advantage function $A^\pi(s, a) = Q^\pi(s, a) - V^\pi(s)$ is the expected increase in reward from selecting $a$ as the next action from state $s$ relative to an action sampled from $\pi$.

**Policy Gradients** Policy gradient methods optimize policy $\pi_\theta$ by directly estimating the gradient of the expected reward with respect to policy parameters. Let $q$ be a prompt, $a$ be the true answer, $\mathbf{o} = (o_1, ..., o_t)$ be a sequence of response tokens, and $\mathrm{r}(\mathbf{o}, a)$ be the final reward received from the verifier. From the policy gradient theorem (Sutton et al., 1999)

$$\hat{g}^{\mathrm{PG}} = \hat{\mathbb{E}}_{\substack{q \sim p(Q), \; a \sim p(a|q), \\ \mathbf{o} \sim \pi_\theta(O|q)}} \left[ \sum_{t=1}^{|\mathbf{o}|} \nabla_\theta \log \pi_\theta(o_t|s_t) \left( \mathrm{r}(\mathbf{o}, a) - b(s_t) \right) \right] \tag{1}$$

is an unbiased estimate of the policy gradient, where $\hat{\mathbb{E}}$ is an empirical sample mean, $s_t := (q, o_{<t})$ is the state at step $t$ (the prompt and prior tokens), and baseline $b(s_t)$ is a function of the current state. A common choice is $b(s_t) = \hat{V}(s_t)$, which makes the baselined reward equivalent to an estimate of the advantage $\hat{A}(s_t, o_t)$. The policy gradient estimator can be interpreted as as increasing the probability

of actions with above average expected rewards and decreasing the probability of actions with below average expected rewards.

Each of the three algorithms considered in this paper (GRPO, PPO, and RLOO) are policy gradient methods. We discuss the different strategies these methods take for advantage estimation and deviations from the policy gradient estimator $\hat{g}^{\text{PG}}$ below.

**Advantage Estimation for Policy Gradients** Consider sampling $G$ responses from a single prompt, and let $\mathbf{r} = (r_1, ..., r_G)$ be the rewards for these responses. Let $\hat{A}_{i,t}$ be the estimated advantage for token $t$ in response $i$. PPO, RLOO, and GRPO then have the following advantage estimators:

| Algorithm | Advantage estimator $\hat{A}_{i,t}$ | Unbiased PG? |
|---|---|---|
| PPO | $r_i - \hat{V}_\psi(s_{i,t})$ | Yes |
| RLOO | $r_i - \text{mean}(\mathbf{r}_{j \neq i})$ | Yes |
| GRPO | $\frac{r_i - \text{mean}(\mathbf{r})}{\text{std}(\mathbf{r}) + \epsilon}$ | No |
| GRPO (No Std Norm) | $r_i - \text{mean}(\mathbf{r})$ | No (proportional) |

PPO uses Generalized Advantage Estimation (GAE) (Schulman et al., 2018) and learns an explicit model of the value function $\hat{V}_\psi$ as a baseline (we focus on the unbiased variant of GAE). To avoid the computational costs associated with learning an explicit value model, RLOO and GRPO instead compute a Monte Carlo estimate of the value using multiple responses generated from the same prompt. Specifically, RLOO subtracts the mean reward from the other sampled responses, yielding an unbiased advantage estimate, while GRPO subtracts the mean reward from all responses and divides by the standard deviation, which is biased. We also consider a variant of GRPO without standard normalization which yields a policy gradient estimate that is proportional to an unbiased estimate (this modification was proposed as part of the Dr. GRPO algorithm (Liu et al., 2025)). We note that RLOO and GRPO uses the same advantage estimate for each token, which can be interpreted as casting question answering as a bandit problem where generating the full response corresponds to a single action.

**Clipped Policy Gradients** The primary contribution of PPO was to introduce a clipped policy gradient estimator to stabilize training when performing multiple gradient updates on a single batch of rollouts (at the cost of introducing bias). The clipped estimator is

$$\hat{g}_t^{\text{clip}} = \nabla_\theta \hat{\mathbb{E}}_{\substack{q \sim p(Q) \\ \mathbf{o} \sim \pi_{\theta_{old}}(O|q)}} \min \left[ \frac{\pi_\theta(o_t|q, o_{<t})}{\pi_{\theta_{old}}(o_t|q, o_{<t})} \hat{A}_t, \text{clip}\left( \frac{\pi_\theta(o_t|q, o_{<t})}{\pi_{\theta_{old}}(o_t|q, o_{<t})}, 1 - \epsilon, 1 + \epsilon \right) \hat{A}_t \right]$$

When applied on-policy, $\pi_\theta = \pi_{\theta_{old}}$ and the clipped estimator reduces to the vanilla policy gradient. The clipped policy gradient is also used in GRPO and can be applied with any of the advantage estimators discussed above.

## 3 EXPERIMENTS

In this section, we empirically compare the performance of GRPO, GRPO without standard normalization, RLOO, and PPO for optimizing language models on probability prediction tasks. Across real-world biological experiments, medical question answering, and synthetic datasets, we find that GRPO induces highly overconfident probability predictions and suboptimal average rewards when optimizing proper-scoring-rule rewards, whereas the other algorithms yield relatively well-calibrated models.

### 3.1 PROBLEM STATEMENT

We study the following **probability prediction task**: given a prompt $q$, predict the conditional probability distribution $p(a \mid q)$ for categorical outcome $a \in \{1, \ldots, K\}$. For example, a model may be queried to predict the probability that each option of a multiple choice question is correct.

To evaluate and optimize the models, we use rewards based on strictly proper scoring rules, which are maximized in expectation only by predicting the true conditional distribution. Let $\hat{\mathbf{p}} = (\hat{p}_1, \ldots, \hat{p}_K)$ be a model's predicted conditional distribution, where $\hat{p}_i = \hat{p}(a = i|q)$. Let $r(\hat{\mathbf{p}}, a)$ be the reward for prediction $\hat{\mathbf{p}}$ and observed outcome $a$. We consider the following rewards (higher is better):

$$\textbf{Log-likelihood:} \qquad r(\hat{\mathbf{p}}, a) = \log \hat{p}_a$$

$$\textbf{Negative Brier score:} \qquad r(\hat{\mathbf{p}}, a) = -\sum_{i=1}^{K} \left(\mathbf{1}[a = i] - \hat{p}_i\right)^2$$

$$\textbf{Spherical score:} \qquad r(\hat{\mathbf{p}}, a) = \frac{\hat{p}_a}{\|\hat{\mathbf{p}}\|_2}$$

Invalid predictions (such as formatting errors or non-normalized probabilities) are assigned a reward equivalent to $\hat{p}_a = 0.01$ for the correct class. Because our tasks have at most four classes, a uniform prediction yields a higher reward under all scoring rules and the models are not incentivized to make invalid predictions.

## 3.2 METRICS

We evaluate models using the following four metrics assessing overall prediction quality, calibration, classification, and ranking:

- **Average reward** — measures overall performance, maximized when predicted probabilities match the true conditional distribution.

- **Expected Calibration Error (ECE)** — quantifies how closely predicted probabilities reflect empirical frequencies by binning predictions (we use $B=10$) and measuring the weighted gap between frequency and predicted probability within each bin. Formally,

$$\text{ECE} = \sum_{j=1}^{M} \frac{|B_j|}{N} \big| \text{Freq}(B_j) - \text{Conf}(B_j) \big|,$$

  where

$$\text{Freq}(B_j) = \frac{1}{|B_j|} \sum_{i \in B_j} a_i, \qquad \text{Conf}(B_j) = \frac{1}{|B_j|} \sum_{i \in B_j} \hat{p}_i,$$

  and $B_j$ denotes the set of examples whose predicted probability falls into bin $j$. In multiclass settings, we use the marginal ECE, treating each class probability and its corresponding binary label as a separate instance.

- **Accuracy** — the fraction of samples for which the class with the highest probability prediction matches the observed outcome.

- **Area Under the Receiver Operating Characteristic (AUROC)** — measures ranking quality for classification, equivalent to probability that a random positive instance has a higher score than a random negative instance. In multiclass settings, We compute AUROC over marginal binary indicators for each class.

## 3.3 EXPERIMENT 1: SCIENTIFIC EXPERIMENT PREDICTION (CRISPR SCREEN)

In this experiment, we compare the performance of each RL algorithm for optimizing language models to reason about the outcome of real-world biological experiments.

**Data** In recent years, Perturb-seq (Dixit et al., 2016) has emerged as a powerful experimental technique for identifying causal effects on cells, a key question in drug discovery, cell engineering, and basic biology research. CRISPR perturb-seq experiments involve perturbing genes with CRISPR and measuring the effect of those perturbations on genome-wide gene expression at single-cell resolution. For this experiment, we convert a large perturb-seq dataset from Replogle et al. (2022) into a binary "hit discovery" task: for a given perturbed gene and target gene expression phenotype,

| Dataset | Algorithm | Qwen3-4B | | | | Llama3.1-8B-Instruct | | | |
|---|---|---|---|---|---|---|---|---|---|
| | | Rwd↑ | ECE↓ | AUROC↑ | Acc↑ | Rwd↑ | ECE↓ | AUROC↑ | Acc↑ |
| CRISPR Screen | GRPO | -1.4 | 0.29 | 0.69 | 0.68 | -1.5 | 0.33 | 0.66 | 0.67 |
| | GRPO (No Std) | -0.6 | 0.04 | 0.72 | 0.71 | -0.7 | 0.13 | 0.71 | 0.65 |
| | RLOO | -0.6 | 0.04 | 0.72 | 0.70 | -0.7 | 0.10 | 0.72 | 0.66 |
| | PPO | -0.6 | 0.04 | 0.72 | 0. | -0.7 | 0.12 | 0.67 | 0.65 |
| Med-MCQA | GRPO | -3.0 | 0.12 | 0.80 | 0.58 | -6.6 | 0.20 | 0.73 | 0.61 |
| | GRPO (No Std) | -1.0 | 0.01 | 0.81 | 0.59 | -1.1 | 0.06 | 0.80 | 0.61 |
| | RLOO | -1.0 | 0.01 | 0.81 | 0.59 | -1.0 | 0.04 | 0.82 | 0.62 |
| | PPO | -1.1 | 0.02 | 0.80 | 0.58 | -1.1 | 0.05 | 0.79 | 0.60 |

Table 1: Evaluation metrics from probability prediction experiments. Across applications to real-world biological experiments and medical question-answering, we find that GRPO achieves poor ECE and AUROC relative to GRPO without standard normalization, RLOO, and PPO. All algorithms perform nearly identically on accuracy with predicted probabilities thresholded at 0.5, which does not require well-calibrated predictions.

predict the probability that the perturbed gene has a strong effect on the phenotype (full preprocessing details in Appendix A.4). Each question effectively asks the model to reason about the probability that a particular causal hypothesis is true. We sample a balanced dataset of positive and negative instances for the final dataset and generate validation and test splits with held-out perturbations.

**Models** We optimize Qwen3-4B (Yang et al., 2025) and Llama3.1-8B-Instruct (Dubey & et al., 2024) for this task. These models represent distinct post-training pipelines (Qwen3-4B has been optimized with RL for chain-of-thought reasoning while Llama3.1-8B-Instruct has not), architectures, and model sizes. Models are prompted to predict the probability between 1% and 99% (full prompts for each model in Appendix A.5).

**Optimization** We optimize the log-likelihood reward defined above with PPO, RLOO, GRPO, and GRPO without standard normalization using Verl (Sheng et al., 2024). Qwen3-4B experiments are conducted with learning rate 1E-6, batch size 512, mini-batch size 64, and maximum response length 2048. In order to reduce memory requirements with a larger model, Llama3.1-8B experiments are conducted with a smaller batch size (128) and maximum response length (1024). All models are trained for 180 steps with the clipped policy gradient. We emphasize that the focus of these experiments is demonstrating the GRPO overconfidence effect and that hyperparameters and that multiple models are included just to show the generality of the effect; hyperparameters have not been extensively tuned to compare the best performance each model can achieve.

**Results** We observe that optimization with GRPO yields substantially worse calibration and average rewards across both the Qwen and Llama models relative to GRPO without standard normalization, RLOO, and PPO (Table 1). Interestingly, models optimized with all algorithms perform similarly on accuracy, though models optimized with GRPO have slightly worse AUROC. We visualize the distribution of predicted probabilities from the Qwen3-4B experiments in 2 and observe that GRPO induces prediction probabilities near 0 or 1 for a vast majority of outcomes. These results are consistent with GRPO optimizing for overconfident yet directionally correct predictions, despite the log-likelihood reward being maximized by predicting the true conditional probability.

**Impact** Calibrated probability predictions are particularly important in scientific experiment planning. Running CRISPR perturbation assays is costly, and decisions such as whether to prioritize, replicate, or further investigate a perturbation can differ if its success probability is 51% or 99%. Beyond practical resource allocation, distinguishing strong from weak evidence is an important capability for robust scientific reasoning.

### 3.4 EXPERIMENT 2: MEDICAL QUESTION ANSWERING (MEDMCQA)

Next, we analyze the behavior of each RL algorithm for medical question answering.

**Data** We use the MedMCQA dataset (Pal et al., 2022), which consists of multiple choice exam questions extracted from medical entrance exams in India. Questions cover diverse tasks and capabilities, including diagnosis, factual recall, and multihop reasoning across medical specialties.

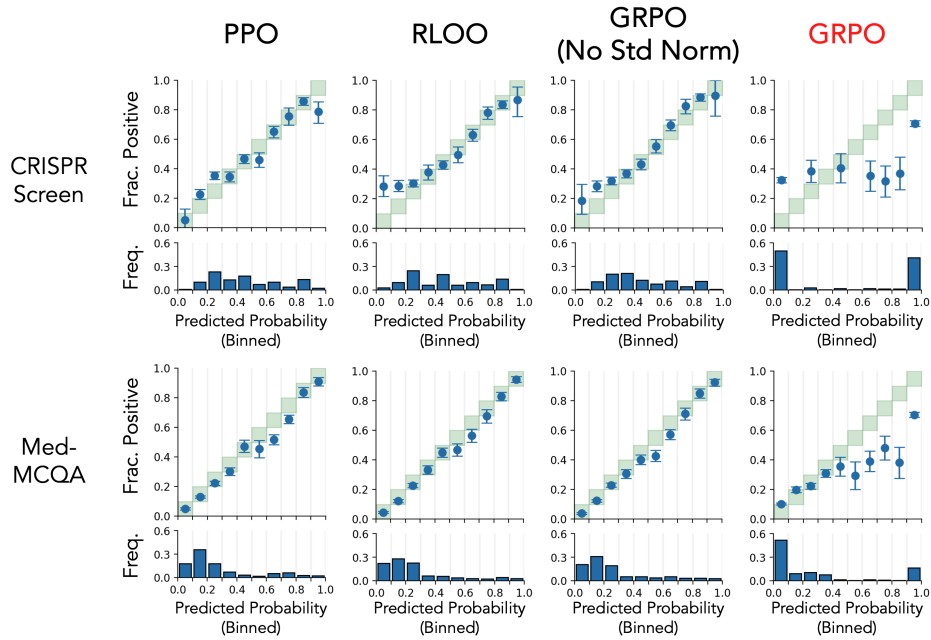

Figure 2: Predictions from the CRISPR screen and MedMCQA experiments after reinforcement learning with Qwen3-4B. In both settings, we observe that models optimized with PPO, RLOO, and GRPO without standard normalization achieve fairly well-calibrated predictions for held-out queries, while models optimized with GRPO make highly overconfident probability predictions. Error bars represent 95% confidence intervals.

Each question has four options. Models are prompted to predict the probability of each multiple choice option (full details Appendix A.6).

**Models and Optimization** We optimize Qwen3-4B and Llama3.1-8B-Instruct using the same RL setup (log-likelihood reward, identical learning rates and batch sizes) as Experiment 1, with the modification of 240 training steps due to the larger dataset.

**Results** We again observe that models optimized with GRPO achieve substantially lower rewards, worse calibration and similar accuracy to models optimized with the other algorithms (Table 1, Figure 2).

**Impact** Calibration is especially important for effective decision making in high-stakes settings like medicine: for example, a highly confident diagnosis may enable quick treatment, and an uncertain diagnosis may motivate further testing and investigation. In this experiment, we see that models optimized with GRPO are incentivized to make highly confident probability predictions even when predictions are often incorrect.

### 3.5 EXPERIMENT 3: SYNTHETIC DATA

The previous experiments demonstrate that GRPO induces overconfidence when optimizing reasoning language models with log-likelihood rewards across two pretrained models and datasets. To examine whether this phenomenon extends beyond the specific models, rewards, and verbalized uncertainty strategy from our prior experiments, we develop a minimal idealized experimental setting with synthetic data. Our results show that GRPO's overconfidence persist across additional scoring rules, models, and uncertainty parameterizations.

**Data** We simulate 10,000 triples $(q_i, c_i, a_i)$ representing questions, categories, and binary answers. Each question is assigned uniformly to one of 20 random categories. For each category $k$, we sample a true Bernoulli parameter $p_k \sim \text{Uniform}(0, 1)$ and answers are drawn as $a_i \sim \text{Bernoulli}(p_{c_i})$.

| Reward | Algorithm | Idealized Token Model | | | | Idealized Probe Model | | | |
|---|---|---|---|---|---|---|---|---|---|
| | | Rwd↑ | ECE↓ | AUROC↑ | Acc↑ | Rwd↑ | ECE↓ | AUROC↑ | Acc↑ |
| Log Likelihood | GRPO | -1.15 | 0.239 | 0.75 | 0.75 | -2.47 | 0.248 | 0.77 | 0.75 |
| | GRPO (No Std) | -0.51 | 0.008 | 0.82 | 0.75 | -0.51 | 0.004 | 0.82 | 0.75 |
| | PPO | -0.51 | 0.007 | 0.82 | 0.75 | -0.51 | 0.012 | 0.82 | 0.75 |
| | RLOO | -0.51 | 0.003 | 0.82 | 0.75 | -0.51 | 0.008 | 0.82 | 0.75 |
| Brier | GRPO | -0.24 | 0.239 | 0.75 | 0.75 | -0.25 | 0.244 | 0.82 | 0.75 |
| | GRPO (No Std) | -0.17 | 0.006 | 0.82 | 0.75 | -0.17 | 0.005 | 0.82 | 0.75 |
| | PPO | -0.17 | 0.009 | 0.82 | 0.75 | -0.17 | 0.016 | 0.82 | 0.75 |
| | RLOO | -0.17 | 0.008 | 0.82 | 0.75 | -0.17 | 0.009 | 0.82 | 0.75 |
| Spherical | GRPO | 0.75 | 0.238 | 0.76 | 0.75 | 0.75 | 0.242 | 0.82 | 0.75 |
| | GRPO (No Std) | 0.81 | 0.008 | 0.82 | 0.75 | 0.81 | 0.006 | 0.82 | 0.75 |
| | PPO | 0.81 | 0.007 | 0.82 | 0.75 | 0.81 | 0.031 | 0.82 | 0.75 |
| | RLOO | 0.81 | 0.010 | 0.82 | 0.75 | 0.81 | 0.014 | 0.82 | 0.75 |

Table 2: Results of synthetic data experiment with a minimal idealized probability prediction models. We find that GRPO consistently yields overconfident probability predictions and poor rewards while still achieving identical maximum likelihood accuracy to other algorithms across model parameterizations and proper scoring rule rewards. AUROC results are sometimes worse due to an increased number of ties as probability predictions approach 0 or 1. These results demonstrate that the GRPO overconfidence phenomenon is not specific to particular pretrained models, architectures, or verbalized uncertainty.

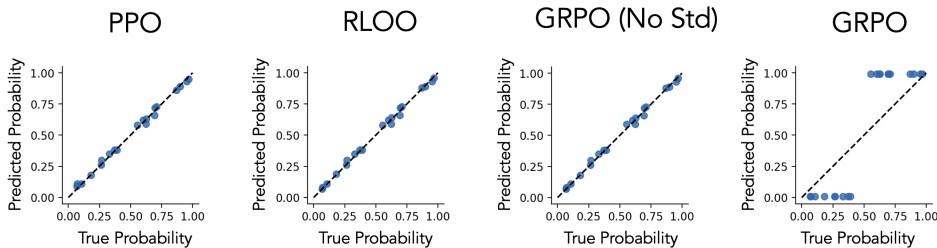

Figure 3: Representative predictions from synthetic data experiment. Models optimized with GRPO converge to predicting probabilities close to one when the true probability is greater than 0.5 and probabilities close to zero when the true probability is less than 0.5. This results in poor calibration and rewards yet strong accuracy of maximum likelihood predictions.

**Model** We design two minimal models for this experiment. The *Idealized Token Model* learns a categorical distribution over 99 discrete probability tokens (0.01-0.99) conditioned on a question's category, mimicking verbalized uncertainty. The *Idealized Probe Model* parameterizes the predicted probability for each category via a beta distribution with learned parameters, mimicking continuous probability heads. For PPO, the value model predicts a single value per category.

**Optimization** We optimize models using GRPO, GRPO without standard normalization, RLOO, and PPO, using group size 5 and 5 gradient updates per rollout. We compute gradients over the full dataset of 10,000 samples. We additionally run an experiment with a very small clipping threshold to measure the impact of gradient clipping. Models are optimized for 40,000 steps with a learning rate of 0.01 with AdamW. Additional details in Appendix **??**

**Results** Across model parameterizations and all three rewards, we find that GRPO yields highly overconfident probability predictions: models optimized with GRPO converge to predict probabilities close to zero for categories with true probability $p_k < 0.5$ and probabilities close to one for categories with true probability $p_k > 0.5$ (Fig. 3). GRPO without standard normalization, PPO, and RLOO all yield well-calibrated predictions (Fig. 3). This pattern is reflected in GRPO's poor rewards and large ECE, despite identical maximum likelihood accuracy across algorithms (Table 2). AUROC is degraded under GRPO due to frequent ties on extreme predictions. We also observe that high

clipping rates do not affect these findings (Tbl. 3, Fig. 7). Taken together, these results show that standard normalization in GRPO's advantage estimate directly drives overconfidence under three proper scoring-rule rewards, independent of model architecture or uncertainty representation.

## 4 THEORETICAL ANALYSIS

Finally, we analyze why standard normalization in GRPO induces overconfident predictions. Recall that GRPO reinforces actions based on their estimated advantage: actions that have large advantages are made more likely, while actions with negative advantages are made less likely. We will show that standard normalization causes GRPO to overestimate the advantage of overconfident predictions, resulting in overconfident policies (Fig. 4).

In Appendix A.1, we derive expressions for the expected advantage estimates from GRPO with and without standard normalization. Let $q$ be a prompt with stochastic answers $a \sim \text{Categorical}(\mathbf{p})$, where $\mathbf{p} = (p_1, ..., p_k)$ are the true conditional probabilities $p(a|q)$. Let $\hat{\mathbf{p}} = (\hat{p}_1, ..., \hat{p}_k)$ be the predicted answer probabilities, and let $\text{r}(\hat{\mathbf{p}}, a)$ be a reward function such as the log-likelihood reward $\text{r}(\hat{\mathbf{p}}, a) = \log \hat{p}_a$. Let $\mu_i = \mathbb{E}_{\hat{\mathbf{p}}' \sim \pi_\theta(\cdot|q)} [\text{r}(\hat{\mathbf{p}}', i)]$ be the expected reward under predictions sampled from the policy if the answer is $i$, and let $\delta_i = \text{r}(\hat{\mathbf{p}}, i) - \mu_i$ be the centered rewards. The true advantage for prediction $\hat{\mathbf{p}}$ is then

$$A(q, \hat{\mathbf{p}}) = \sum_{i=1}^{k} p_i \delta_i$$

Next, we consider the advantage estimate for GRPO without standard normalization. Let $\hat{\mathbf{p}}^{(1)}, ..., \hat{\mathbf{p}}^{(G)} \sim \pi_\theta(\cdot \mid q)$ be a group of $G$ i.i.d. predictions sampled from prompt $q$, and let $\delta_i^{(j)} = \text{r}(\hat{\mathbf{p}}^{(j)}, i) - \mu_i$ be the centered rewards from prediction $j$ if the answer is $i$. Without loss of generality, we consider estimating the expected advantage for the first prediction in the group. We show that the expected advantage estimate for GRPO without standard normalization, averaging over answers and other predictions in the group, is

$$\mathbb{E}_{a, \hat{\mathbf{p}}^{(2:G)}} \left[ \hat{A}^{\text{NO-STD}}(q, \hat{\mathbf{p}}^{(1)}) \right] = \frac{G-1}{G} \sum_{i=1}^{k} p_i \delta_i^{(1)} = \frac{G-1}{G} A(q, \hat{\mathbf{p}}^{(1)})$$

This means that the policy gradients using GRPO without standard normalization are approximately unbiased (up to a constant factor), consistent with the calibrated predictions we observed experimentally.

Next, we assess the expected advantage estimate for GRPO with standard normalization. Let $\hat{\sigma}_i = \text{std}(\text{r}(\hat{\mathbf{p}}^{(1)}, i), ..., \text{r}(\hat{\mathbf{p}}^{(G)}, i))$ be the standard deviation of rewards within a group. We show that the expected advantage estimate for GRPO is then:

$$\mathbb{E}_{a, \hat{\mathbf{p}}^{(2:G)}} \left[ \hat{A}^{\text{STD}}(q, \hat{\mathbf{p}}^{(1)}) \right] = \frac{G-1}{G} \sum_{i=1}^{k} p_i \, \delta_i^{(1)} \underbrace{\mathbb{E}_{\hat{\mathbf{p}}^{(2:G)}} \left[ \frac{1}{\hat{\sigma}_i + \epsilon} \right]}_{\substack{\text{Inverse-Variance} \\ \text{Amplification}}} - \frac{1}{G} \sum_{i=1}^{k} p_i \, \underbrace{\text{Cov}_{\hat{\mathbf{p}}^{(2:G)}} \left( \sum_{j=2}^{G} \delta_i^{(j)}, \frac{1}{\hat{\sigma}_i + \epsilon} \right)}_{\text{Covariance Bias}}$$

We see that standard normalization introduces two policy-dependent biases to the GRPO advantage estimate: an inverse-variance amplification term, which amplifies reward deviations when the reward variance is small, and a covariance bias term, which captures the dependence between the group mean reward and its variability.

To illustrate the effect of these policy-dependent biases, we estimate the expected advantage of GRPO with and without standard normalization through simulation (Figure 4). We consider a binary setting with outcomes drawn from Bernoulli(0.7) and evaluate advantage estimates from three policies: a uniform policy, a policy concentrated on the true conditional probability, and a policy concentrated on highly overconfident predictions. We sample outcomes and predictions from each policy with group

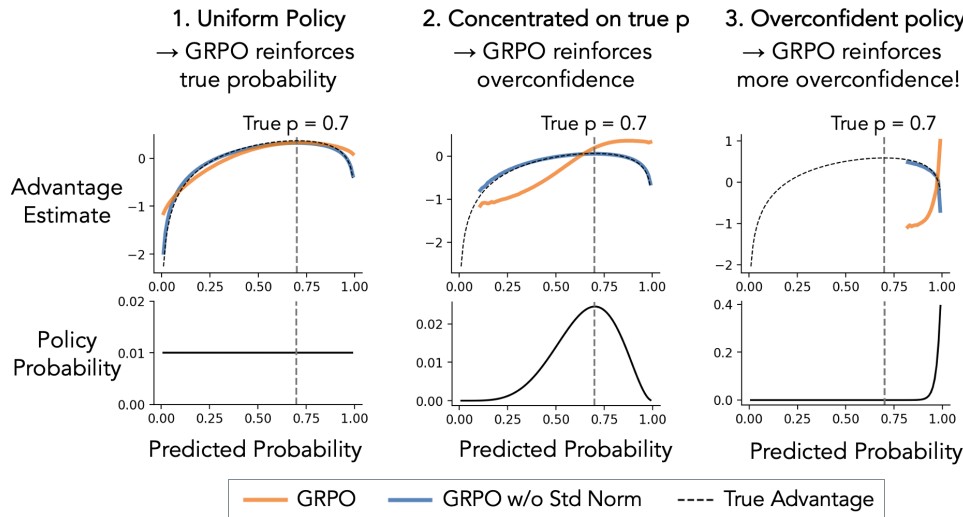

Figure 4: Bias in GRPO advantage estimates explains overconfident predictions. Advantages are computed with a log-likelihood reward. **Left:** Under a uniform policy, both GRPO and GRPO without standard normalization closely approximate the true advantages. **Middle:** Under a policy concentrated on the true probability, GRPO overestimates the advantage of overconfident predictions. **Right:** As the policy becomes overconfident, GRPO increasingly overestimates the advantage overconfident predictions. This pattern creates a positive feedback loop towards increasingly overconfident predictions consistent with our experimental observations.

size $G = 8$ and estimate the expected advantage with a log-likelihood reward for each predicted probability. Under a uniform policy, both estimators are accurate (left column). As the policy begins to concentrate around the true probability, we observe that GRPO overestimates the advantage of overconfident predictions, while the unnormalized advantage estimate remains accurate (center column). This causes GRPO to reinforce overconfident predictions more strongly than the true probability, resulting in overconfident policies. Finally, with a very overconfident policy, GRPO's advantage estimates increasingly overestimate the advantage of overconfident predictions, while GRPO without standard normalization remains approximately unbiased (right column).

To summarize, group standard normalization in GRPO's advantage estimates creates a policy-dependent bias that pushes policies towards overconfident predictions. We observe a consistent pattern with other proper scoring rule rewards (Appendix A.2).

## 5 RELATED WORK

**Calibration of Language Models** Prior work documents that large language models are often well calibrated after pretraining but become overconfident after reinforcment learning from human feedback (RLHF) finetuning with algorithms like PPO or direct preference optimization (DPO) (Kadavath et al., 2022; OpenAI & et al., 2024). We note that this does not contradict our experiments showing calibrated predictions following optimization with PPO: unlike the strictly proper scoring rule rewards we study, RLHF rewards are not necessarily maximized by predicting calibrated probabilities.

Beyond token-level probabilities, prior works demonstrate overconfidence in uncertainty elicited via verbalized estimates and sampling-based strategies through natural language (verbalized uncertainty) and via sampling (Tian et al., 2023; Xiong et al., 2023). Tian et al. (2023) find that verbalized confidence remains better calibrated than token logits in RLHF-trained models. Mei et al. (2025) further show that recent reasoning models also exhibit substantial overconfidence.

Given these issues, several approaches have been developed to improve the calibration of language models. Kadavath et al. (2022) show that simple temperature scaling (Guo et al., 2017) improves

calibration on multiple-choice QA. Mielke et al. (2022) aimed to improve verbalized uncertainty estimates by fitting a probe model to predict confidence from internal representations and adjusting outputs based on the predicted confidence. While post-hoc methods may be effective, they are limited in the reasoning setting where overconfident intermediate claims can pose challenges for compositional reasoning and generalization. More relevant to our work are RL-based methods that directly optimize proper scoring rules. Band et al. (2024) apply proper scoring rule rewards for uncertainty estimation with long-form, multi-claim responses and demonstrate improved calibration across tasks with non-reasoning models. Xu et al. (2024) and Stangel et al. (2025) similarly optimize proper scoring rules for calibration in non-reasoning models. Concurrent with our work, Damani et al. (2025) propose blending accuracy with Brier score to train calibrated reasoning models.

**GRPO Variants** Since the introduction of GRPO (Shao et al., 2024), numerous variants have aimed to simplify or debias the algorithm. A key line of work proposes removing the standard-deviation normalization term, including Dr. GRPO (Liu et al., 2025) and Group Policy Gradient (GPG) (Chu et al., 2025). However, these papers do not demonstrate the negative impact of standard normalization: Dr. GRPO describes a potential bias in gradient weighting based on question difficulty, and GPG identifies a risk of distorted rewards, but neither demonstrates improved performance on the accuracy task they consider by removing the term.

Our work differs in both setting and conclusion. We show that when applied to strictly proper scoring rule rewards, standard normalization causes substantial miscalibration and degraded reward, and that removing normalization yields markedly better-calibrated models. This highlights an important failure mode of GRPO that only shows up outside the standard accuracy-focused setting and which can inform future algorithm development and selection.

**GRPO Overconfidence** During review, we were made aware of Turtel et al. (2025), which was published shortly before our work and studies reasoning models trained for forecasting with a Brier reward. In their experiments, they independently observe that GRPO produces overconfident predictions and that removing standard normalization reduces the proportion of extreme predictions (from 39% to 13% outside the 10–90% range) and slightly improves ECE (0.21 to 0.20 from removing standard normalization in their experiment).

Our work complements theirs by providing a broader and deeper analysis of how and why standard normalization in GRPO drives overconfidence. We show that GRPO yields severe miscalibration and suboptimal rewards across multiple datasets, scoring rules, and model families; that GRPO-trained models surprisingly attain comparable accuracy to better-calibrated alternatives; and provide a theoretical characterization how overconfidence arises from group standard normalization . Our results provide a more complete characterization of how biased advantage estimates from standard normalization in GRPO derail optimization when applied to settings beyond accuracy, an important consideration for future methods.

## 6 DISCUSSION

Many important tasks, from scientific experimentation to uncertainty estimation, require reasoning about the probability of uncertain outcomes. We showed that GRPO induces highly overconfident predictions when optimizing three proper scoring rule rewards, whereas PPO and RLOO are effective at optimizing calibrated probability estimates. We further identified group standard normalization as the source of GRPO's bias, provided a theoretical explanation for why this normalization drives overconfidence, and confirmed that PPO's clipped policy gradient does not drive miscalibration in our experiments.

These findings inform RL algorithm selection in two ways. Most narrowly, they provide evidence against using GRPO with standard normalization for probability estimation tasks. More broadly, they highlight the risks of applying algorithms with biased advantage estimates to new objectives, where such biases may misalign optimization. Finally, our results underscore an important design principle for future RL methods: while unbiased advantage estimates provide a consistent optimization signal across both deterministic and probabilistic tasks, biased estimators must be carefully aligned with the structure of the target objective to be effective.

## 7 REPRODUCIBILITY STATEMENT

The code and data required to reproduce experiments and figures are provided in supplementary materials.

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

# A APPENDIX

## A.1 ANALYSIS OF BIAS IN GRPO ADVANTAGE ESTIMATES

Let $q$ be a prompt with stochastic answers $a \sim \text{Categorical}(\mathbf{p})$, where $\mathbf{p} = (p_1, ..., p_k)$ are the true answer probabilities. Let $\hat{\mathbf{p}} = (\hat{p}_1, ..., \hat{p}_k)$ be the predicted answer probabilities from policy $\pi_\theta$. Let $\text{r}(\hat{\mathbf{p}}, a)$ be a reward function from a strictly proper scoring rule such as the log-likelihood reward $\text{r}(\hat{\mathbf{p}}, a) = \log \hat{p}_a$. Rewards based on strictly proper scoring rules have the property that the expected value is maximized by predicting the true conditional probability and they have been shown to be effective rewards for training calibrated classifiers (Band et al., 2024).

Let $\mu_i = \mathbb{E}_{\hat{\mathbf{p}}' \sim \pi_\theta(\cdot|q)} \left[ \text{r}(\hat{\mathbf{p}}', i) \right]$ be the expected reward under predictions sampled from the policy if the answer is set to be $i$, and let $\delta_i = \text{r}(\hat{\mathbf{p}}, i) - \mu_i$ be the centered reward. Then the **true advantage** for prompt $q$ and prediction $\hat{\mathbf{p}}$ is

$$
\begin{aligned}
A(q, \hat{\mathbf{p}}) &= Q^\pi(q, \hat{\mathbf{p}}) - V^\pi(q) \\
&= \mathbb{E}_{a \sim p(a|q)}[\text{r}(\hat{\mathbf{p}}, a)] - \mathbb{E}_{a \sim p(a|q), \hat{\mathbf{p}}' \sim \pi_\theta(\cdot|q)}[\text{r}(\hat{\mathbf{p}}', a)] \\
&= \sum_{i=1}^{k} p_i \text{r}(\hat{\mathbf{p}}, i) - \mathbb{E}_{\hat{\mathbf{p}}' \sim \pi_\theta(\cdot|q)} \sum_{i=1}^{k} p_i \text{r}(\hat{\mathbf{p}}', i) \\
&= \sum_{i=1}^{k} p_i (\text{r}(\hat{\mathbf{p}}, i) - \mu_i) \\
&= \sum_{i=1}^{k} p_i \delta_i
\end{aligned}
$$

Next, we compare the advantage estimates from GRPO (Shao et al., 2024) to the true advantage to characterize any biases. Let $\hat{\mathbf{p}}^{(1)}, ..., \hat{\mathbf{p}}^{(G)} \sim \pi_\theta(\cdot \mid q)$ be a group of $G$ i.i.d. predictions sampled conditional on prompt $q$. Without loss of generality, we will set the index of the prediction whose advantage we are estimating to be $1$. The expected advantage for GRPO without standard normalization is

$$
\begin{aligned}
\mathbb{E}_{\substack{a \sim p(a|q), \\ \hat{\mathbf{p}}^{(2:G)} \sim \pi_\theta(\cdot|q)}} \left[ \hat{A}^{\text{NO-STD}}(q, \hat{\mathbf{p}}^{(1)}) \right] &= \mathbb{E}_{a, \hat{\mathbf{p}}^{(2:G)}} \left[ \text{r}(\hat{\mathbf{p}}^{(1)}, a) - \frac{1}{G} \sum_{j=1}^{G} \text{r}(\hat{\mathbf{p}}^{(j)}, a) \right] \\
&= \mathbb{E}_{a, \hat{\mathbf{p}}^{(2:G)}} \left[ \left(1 - \frac{1}{G}\right) \text{r}(\hat{\mathbf{p}}^{(1)}, a) - \frac{1}{G} \sum_{j=2}^{G} \text{r}(\hat{\mathbf{p}}^{(j)}, a) \right] \\
&= \frac{G-1}{G} \mathbb{E}_a \left[ \text{r}(\hat{\mathbf{p}}^{(1)}, a) - \mu_a \right] \\
&= \frac{G-1}{G} \sum_{i=1}^{k} p_i \delta_i \\
&= \frac{G-1}{G} A(q, \hat{\mathbf{p}}^{(1)})
\end{aligned}
$$

The advantage estimate is proportional to the true advantage, as shown above, though it is attenuated by a factor of $\frac{G-1}{G}$. A fully unbiased estimate can be achieved with the advantage from RLOO (Kool et al., 2019), which excludes $\hat{\mathbf{p}}^{(1)}$ from the mean baseline.

Finally, we consider the expected advantage estimate for GRPO with standard normalization. For notational convenience, let $r_i^{(j)} := \text{r}(\hat{\mathbf{p}}^{(j)}, i)$, $\delta_i^{(j)} := r_i^{(j)} - \mu_i$, and define $\hat{\sigma}_i = \text{std}(r_i^{(1)}, \ldots, r_i^{(G)})$. Taking the expectation over answers and all predictions other than the first, we have

$$\mathbb{E}_{\substack{a \sim p(a|q), \\ \hat{\mathbf{p}}^{(2:G)} \sim \pi_\theta(\cdot|q)}} \left[ \hat{A}^{\mathrm{STD}}(q, \hat{\mathbf{p}}^{(1)}) \right] = \mathbb{E}_{a, \hat{\mathbf{p}}^{(2:G)}} \left[ \frac{r_a^{(1)} - \frac{1}{G} \sum_{j=1}^{G} r_a^{(j)}}{\hat{\sigma}_a + \epsilon} \right]$$

$$= \mathbb{E}_{\hat{\mathbf{p}}^{(2:G)}} \sum_{i=1}^{k} p_i \left[ \frac{r_i^{(1)} - \frac{1}{G} \sum_{j=1}^{G} r_i^{(j)}}{\hat{\sigma}_i + \epsilon} \right]$$

$$= \mathbb{E}_{\hat{\mathbf{p}}^{(2:G)}} \sum_{i=1}^{k} p_i \left[ \frac{\frac{G-1}{G} r_i^{(1)} - \frac{1}{G} \sum_{j=2}^{G} r_i^{(j)}}{\hat{\sigma}_i + \epsilon} \right]$$

$$= \mathbb{E}_{\hat{\mathbf{p}}^{(2:G)}} \sum_{i=1}^{k} p_i \left[ \frac{\frac{G-1}{G} \left( r_i^{(1)} - \mu_i \right) - \frac{1}{G} \sum_{j=2}^{G} (r_i^{(j)} - \mu_i)}{\hat{\sigma}_i + \epsilon} \right]$$

$$= \frac{G-1}{G} \sum_{i=1}^{k} p_i \, \delta_i^{(1)} \, \mathbb{E}_{\hat{\mathbf{p}}^{(2:G)}} \left[ \frac{1}{\hat{\sigma}_i + \epsilon} \right] - \frac{1}{G} \sum_{i=1}^{k} p_i \, \mathbb{E}_{\hat{\mathbf{p}}^{(2:G)}} \left[ \frac{\sum_{j=2}^{G} \delta_i^{(j)}}{\hat{\sigma}_i + \epsilon} \right]$$

We note that the expectation of the centered rewards with respect to predictions is zero: $\mathbb{E}_{\hat{\mathbf{p}}^{(j)}}[\delta_i^{(j)}] = \mathbb{E}_{\hat{\mathbf{p}}^{(j)}}[r_i^{(j)}] - \mu_i = 0$. Using $\mathbb{E}[XY] = \mathbb{E}[X]\mathbb{E}[Y] + \mathrm{Cov}(X, Y)$, we find

$$\mathbb{E}_{\hat{\mathbf{p}}^{(2:G)}} \left[ \frac{\sum_{j=2}^{G} \delta_i^{(j)}}{\hat{\sigma}_i + \epsilon} \right] = \mathrm{Cov} \left( \sum_{j=2}^{G} \delta_i^{(j)}, \frac{1}{\hat{\sigma}_i + \epsilon} \right).$$

This yields our final expression for the expected advantage estimate for GRPO with standard normalization:

$$\mathbb{E}_{a, \hat{\mathbf{p}}^{(2:G)}} \left[ \hat{A}^{\mathrm{STD}}(q, \hat{\mathbf{p}}^{(1)}) \right] = \frac{G-1}{G} \sum_{i=1}^{k} p_i \, \delta_i^{(1)} \underbrace{\mathbb{E} \left[ \frac{1}{\hat{\sigma}_i + \epsilon} \right]}_{\substack{\text{Inverse-Variance} \\ \text{Amplification}}} - \frac{1}{G} \sum_{i=1}^{k} p_i \underbrace{\mathrm{Cov} \left( \sum_{j=2}^{G} \delta_i^{(j)}, \frac{1}{\hat{\sigma}_i + \epsilon} \right)}_{\text{Covariance Bias}}$$

where the expectations in the final expression are over the group predictions other than $\hat{\mathbf{p}}^{(1)}$. We observe two policy dependent biases: inverse variance term that multiplies the centered reward terms seen in the GRPO no-standard advantage and a covariance bias introduced by the dependence of group reward means and standard deviations. We analyze the effect of these biases empirically in Figures 4, 5, and 6.

## A.2 GRPO BIAS WITH OTHER REWARDS

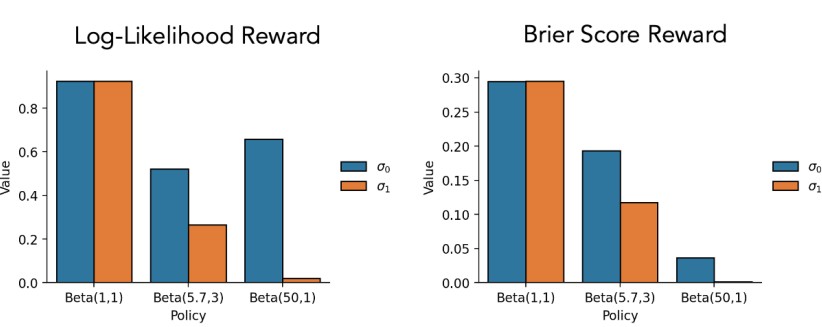

Figure 5: Analysis of advantage estimates with a reward based on the Brier score. We observe a similar pattern of overestimated advantages for overconfident probabilities as observed with a log-likelihood in Fig. 4

.

Figure 6: Empirical estimates of $\sigma_0$ and $\sigma_1$ (standard deviation of rewards within groups for answers 0 and 1) for the three policies in Figures 4 and 5. As the policies concentrate on predictions greater than 0.5, $\sigma_0$ becomes larger than $\sigma_1$.

A.3    SYNTHETIC DATA EXPERIMENT EXTENDED RESULTS

| Algorithm | Grad Steps / Rollout | $\epsilon_{\text{clip}}$ | ECE | AUROC | Accuracy |
|---|---|---|---|---|---|
| GRPO | 1 | NA | 0.239 | 0.750 | 0.751 |
| GRPO | 10 | 0.200 | 0.239 | 0.751 | 0.751 |
| GRPO | 10 | 0.001 | 0.239 | 0.751 | 0.751 |
| GRPO (No Std) | 1 | NA | 0.002 | 0.823 | 0.751 |
| GRPO (No Std) | 10 | 0.200 | 0.005 | 0.823 | 0.751 |
| GRPO (No Std) | 10 | 0.001 | 0.005 | 0.823 | 0.751 |
| PPO | 1 | NA | 0.005 | 0.823 | 0.751 |
| PPO | 10 | 0.200 | 0.008 | 0.823 | 0.751 |
| PPO | 10 | 0.001 | 0.008 | 0.823 | 0.751 |
| RLOO | 1 | NA | 0.002 | 0.823 | 0.751 |
| RLOO | 10 | 0.200 | 0.004 | 0.823 | 0.751 |
| RLOO | 10 | 0.001 | 0.004 | 0.823 | 0.751 |

Table 3: Extended results from synthetic data experiments. We analyze performance online and offline (1 and 10 steps per rollout) using the idealized token model and log-likelihood rewards. We observe that results are consistent between experiments with a single update per rollout and multiple updates per rollout with a clipped policy gradient estimates, even with low clipping thresholds that encourage high clipping rates.

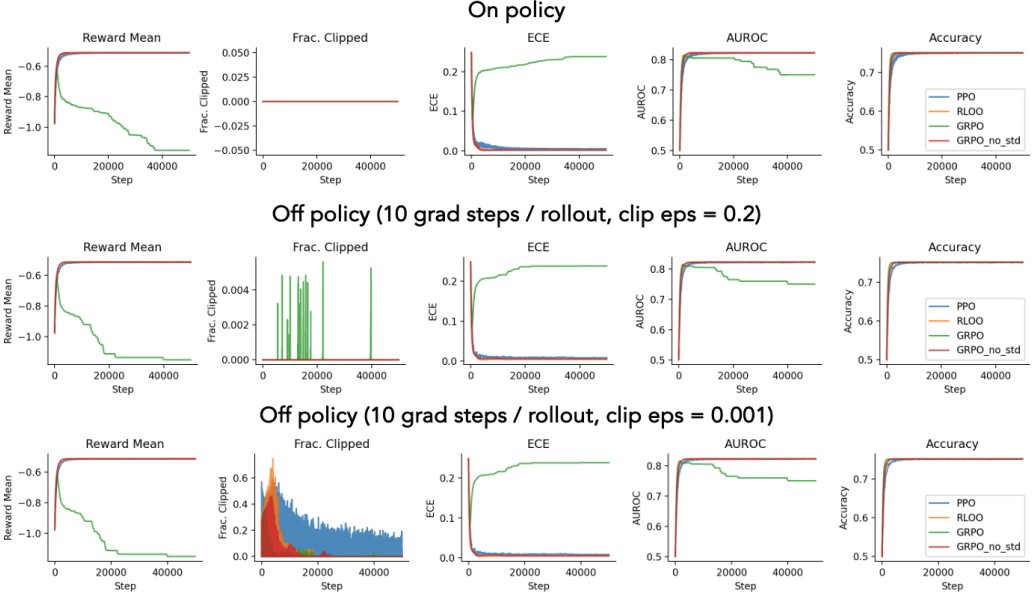

Figure 7: Extended synthetic data experiment metrics. We note that rewards are consistent even with high clipping rates.

### A.4 CRISPR EXPERIMENT DATA PROCESSING

CRISPR perturb-seq screens involve perturbing individual genes with CRISPR (which modulates the expression of a target gene) and measuring the effect of the perturbation on RNA transcript counts for all genes in individual cells cell. We use the essential gene CRISPRi (CRISPR interference) perturb-seq screen in K562 cells from Replogle et al. (2022) for our experiment. The dataset contains CRISPRi perturbations, which lower gene expression, that target approximately 2,000 unique genes. We apply consensus non-negative matrix factorization (cNMF) (Kotliar et al., 2019) to infer 50 aggregate transcriptional target phenotypes and select the top 15 marker genes for each phenotype as defined by the cNMF method to describe each phenotype. We estimate the effect size of each

perturbation on each phenotype as the difference in mean phenotype values for cells that received the perturbation and control cells. To define perturbations with strong effects ("hits"), we fit a cluster model on the perturbation effect sizes for each phenotype and select perturbations that are highly unlikely in the control cluster. Specifically, we fit a Gaussian Mixture Model on the effect sizes for each phenotype (number of clusters between 1-4, selected based on Bayesian Information Criterion) and select perturbations with $<1\%$ chance under the cluster closest to zero as strong effects. To construct a balanced dataset, we select an equal number of perturbations that are most likely under the control cluster as non-hits. We note that the dataset is naturally very imbalanced (hits are relatively rare for most phenotypes) but choose to work with a balanced dataset for simplicity as our primary focus is understanding the behavior of RL algorithms with stochastic outcomes.

## A.5 CRISPR TASK PROMPT

---

**CRISPR Screen Prompt (Qwen3-4B)**

```
I am planning a perturb-seq screen and plan to assess effects of
↪   perturbations on a phenotype with the following marker genes:
↪   ```{pheno_markers}```.

How likely is a CRISPRi perturbation applied to {pert} to have a
↪   strong effect on this phenotype? Respond with probability from
↪   1-99, representing 1% to 99% chance of a strong effect. Enclose
↪   your answer in <answer> </answer> tags.
```

---

**CRISPR Screen Prompt (Llama3.1-8B-Instruct)**

```
I am planning a perturb-seq screen and plan to assess effects of
↪   perturbations on a phenotype with the following marker genes:
↪   ```{pheno_markers}```.

How likely is a CRISPRi perturbation applied to {pert} to have a
↪   strong effect on this phenotype? This perturbation was sampled
↪   from a balanced dataset where 50% of perturbation / phenotype
↪   pairs have strong effects.

Respond with probability from 1-99, representing 1% to 99% chance
↪   of a strong effect. Your goal is to use your biological
↪   knowledge to distinguish likely vs unlikely effects, so predict
↪   above 50% if there is reason to believe an effect is more
↪   likely than average or less that 50% if it is less likely than
↪   average.

Concisely reason about the probability of a strong effect, then
↪   output your final answer enclosed in <answer> </answer> tags.

Example output: [concise reasoning about the probability]
↪   <answer>84</answer>
```

---

We use a separate prompt for the Llama model to encourage reasoning and reduce the rate of refusals. We emphasize that our experiments are designed to compare each algorithm and demonstrate the GRPO overconfidence phenomenon, not optimize or match hyperparameters exactly to compare models.

`pheno_markers` is a list of 15 marker genes for the phenotype, and `pert` is the gene perturbed by the CRISPR perturbation. We also considered prompts that specified the overall frequency of hits in the dataset, but found that this reduced the zero-shot model performance.

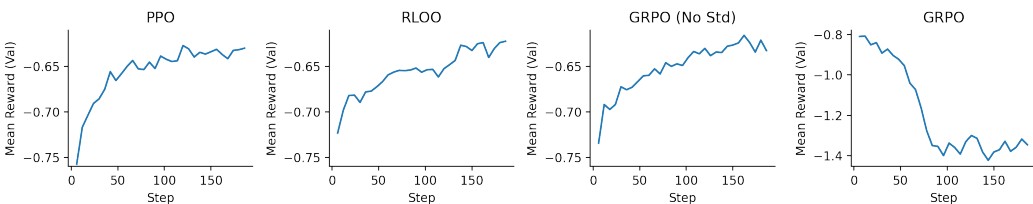

Figure 8: Qwen3-4B CRISPR experiment prediction task validation set rewards during training.

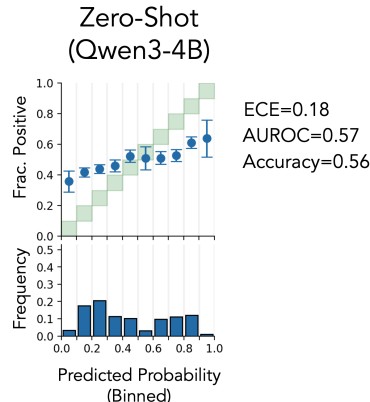

Figure 9: Zero shot predictions on CRISPR task test set with Qwen3-4B.

### A.6 MedMCQA Task Prompt

```
MedMCQA Prompt (Qwen3-4B)

Predict the probability that each answer to the following multiple
↪   choice question is correct. Reason carefully about the
↪   probability of each answer, and make sure to be precise--if
↪   possible, predict the exact probability.

Formatting: Enclose your answer with <answer> <\answer> tags. Write
↪   each probability with up to two decimal places. None of your
↪   predictions can have value exactly equal to 0 or 1. Format your
↪   answer as a comma separated list for the probability of options
↪   A,B,C,D. For example, a valid answer is
↪   `<answer>0.28,0.61,0.04,0.07</answer>`.

Reminders: Your probabilities must sum to 1, and 0.00 and 1.00 are
↪   not valid responses!

Question: {question}

A: {opa}
B: {opb}
C: {opc}
D: {opd}
```

---

**MedMCQA Prompt (Llama3.1-8B-Instruct)**

```
Predict the probability that each answer to the following multiple
↪   choice question is correct. Reason carefully about the
↪   probability of each answer, and make sure to be precise--if
↪   possible, predict the exact probability.

Formatting: Concisely reason step-by-step about your answer. Then
↪   write your final answer enclosed in <answer> <\answer> tags.
↪   Write each probability with up to two decimal places. None of
↪   your predictions can have value exactly equal to 0 or 1. Format
↪   your answer as a comma separated list for the probability of
↪   options A,B,C,D.

Example output: [concise reasoning about the correct answer and
↪   answer probabilities] <answer>0.05,0.12,0.79,0.04</answer>

Reminders: Your probabilities must sum to 1, and 0.00 and 1.00 are
↪   not valid responses!

Question: {question}

A: {opa}
B: {opb}
C: {opc}
D: {opd}
```

---

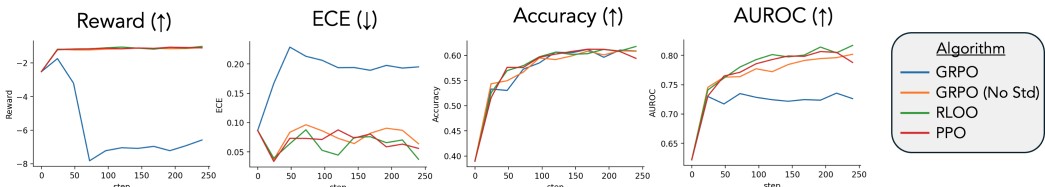

Figure 10: Probability prediction validation statistics for MedMCQA Llama3.1-8B-Instruct run. We observe that the model log-likelihood reward and expected calibration error quickly deteriorate when optimizing predictions with GRPO, though the model still achieves comparable accuracy and slightly lower AUROC to models optimized with other methods.

## B  LLM USAGE

The most prominent usage of LLM's in writing this paper was to search for related literature (request ChatGPT to find papers related to specific topics). ChatGPT was also used to as a general purpose search and question-answering tool, for example to for latex formatting and to refine mathematical derivations and notation, and as a tool for revising writing.

