# OpenReview forum: "Uncalibrated Reasoning: GRPO Induces Overconfidence for Stochastic Outcomes"
_ICLR.cc/2026/Conference — Submitted to ICLR 2026_

### Official Review · Reviewer_GDzm · 2025-10-19

**Soundness:** 2
**Presentation:** 1
**Contribution:** 2
**Rating:** 2
**Confidence:** 4

**Summary:**

The paper claims that standard GRPO makes language models become overconfident, and compares with PPO and RLOO with several experiments. Emprically, it shows that on multiple datasets, GRPO creates overconfidence, while PPO, RLOO, and GRPO w/o std norm can keep model calibration. Theoretically, it shows that the expected average of standard GRPO is biased, and creates a positive feedback loop that pushes the model to become more extreme in its predictions.

**Strengths:**

- The paper shows a clear experiment result that GRPO causes model overconfidence on three different datasets.
- They show that a simple fix can be done by removing the standard‑normalization in GRPO.
- They also give a theoretical explanation that standard GRPO's advantage estimation biases updates toward overconfident predictors.

**Weaknesses:**

- The main problem is that this paper does not include a related work section. There is a lack of discussion of previous literature on model calibration, different post-training methods, etc.
- In fact, many previous papers have pointed out that other post-training methods (PPO, DPO) can also lead to model overconfidence, which conflicts with this paper's experiments. The authors do not discuss these relevant works and their connections with this paper.
- The experiment lacks enough generality. The authors only experiment with Qwen3-4B, without considering other architectures, such as OctoThinker based on Llama models. Dataset-wise, the authors only use three uncommon datasets for evaluation, without reporting on reasoning benchmarks such as MATH, AIME-24/25, etc.

**Questions:**

How sensitive are results to group size G? Your theoretical analysis assumes a large group size G, but you use a small G=4 in your CRISPR experiments.

---

> ### Author Response · Authors · 2025-12-04
>
> _1. Need for further discussion of related work_
>
> Thank you for the suggestion! We agree that a deeper discussion of related work helps position and clarify our contributions. We have added a related work section that discusses prior work to measure and calibrate LLMs and post-training method variants.
>
> _2. Relationship to prior overconfidence observed with PPO and DPO in RLHF_
>
> Thanks for the question! We believe there is no conflict between our results (PPO yields calibrated probability estimates when optimizing a proper scoring rule) and prior work observing overconfidence in models that have undergone RLHF with PPO / DPO [1]. The key difference is the reward being optimized: in RLHF, the reward is (a model of) user preference, which is not necessarily maximized via calibrated predictions. In our case, we are optimizing objectives that are maximized only by predicting calibrated probabilities. In fact, prior work [2] has shown that PPO can be used to optimize LLMs for calibration in a non-reasoning setting!
>
> This is a great point to clarify and we have added a discussion in the related work section.
>
> _3. Experiments lack generality_
>
> Thanks for the suggestions! We have added experiments with Llama3.1-8B-Instruct, which represents a different model scale, architecture, and post-training pipeline than Qwen3-4B, and observe the same pattern of overconfidence with GRPO. While understanding the performance of uncertainty estimation on mathematical tasks in particular could be interesting, we consider beyond the scope of our work. Our objective is to demonstrate that GRPO yields overconfident predictions and poor rewards when optimizing probability predictions based on proper scoring rules. We believe that our empirical experiments spanning multiple datasets, model architectures, and scoring functions, along with our theoretical analysis linking GRPO's biased advantage estimates with overconfidence, provide strong evidence that this effect is robust.
>
> _4. Question regarding sensitivity to group size_
>
> Thanks for the great question! We address this with a couple updates:
>
> a ) Update our expression theoretical analysis to no longer require large group sizes
>
> b) Update the visualization of empirical advantage estimates to use a small group size (G=8), which qualitatively exhibits the same behavior as the empirical estimates with a large group size
>
> Thanks again for the valuable feedback!
>
> [1] GPT4 Technical Report. arxiv 2024
>
> [2] Linguistic Calibration of Long-Form Generations. ICML 2024

---

### Official Review · Reviewer_2txt · 2025-10-27

**Soundness:** 2
**Presentation:** 1
**Contribution:** 2
**Rating:** 2
**Confidence:** 4

**Summary:**

The authors claim that Group Relative Policy Optimization (GRPO) produces overconfident probability estimates when trained with a log-likelihood reward, whereas PPO and RLOO yield well-calibrated predictions.

Experiments are conducted on synthetic probability prediction, a CRISPR biological dataset, and the MedMCQA medical question-answering dataset. Across all tasks, GRPO models show poor calibration (high ECE), even though classification accuracy remains similar. The authors attribute this issue to the group standard normalization in GRPO’s advantage estimate, arguing theoretically that it introduces policy-dependent bias amplifying overconfidence.

**Strengths:**

1. Interesting empirical observation. The finding that GRPO causes overconfidence in stochastic prediction tasks is novel and may interest researchers exploring calibration or uncertainty in RL algorithms.

2. Practical implication. The fix—removing standard normalization—is simple and easy to test. If correct, it could inform best practices for future reasoning RL work.

3. Link theory and practice. The discussion about group normalization introducing a bias in the advantage estimate points toward a principled mechanism, not just an empirical finding.

**Weaknesses:**

1. Limited originality and contribution. The main claim—removing normalization from GRPO improves behavior—closely parallels Dr. GRPO [1], which already proposed removing the same term. The paper essentially re-validates known ideas rather than introducing a distinct algorithmic or theoretical innovation. Calibration of GRPO is certainly a bug worth fixing, but the contribution is incremental—basically a note explaining why an already known fix works—rather than delivering a substantial new methodological or conceptual advance.

2. Shallow theoretical analysis. The “theoretical explanation” seems mostly heuristic: it sketches the bias term but does not mathematically prove the direction or magnitude of the effect. Many simplifying assumptions severely undermine rigor.

3. Weak experimental design. The synthetic dataset is a toy-like one and nearly trivial; calibration differences there do not convincingly extend to real settings. The Qwen 3‑4B experiments use very small training sizes, few seeds, and no standard deviation reporting. Metrics are reported without statistical significance or uncertainty intervals.

4. Writing and presentation issues. The paper is poorly organized, with the main content spanning only seven and a half pages; it is more like a lab note rather than a polished paper.

[1] Liu, Zichen, et al. "Understanding r1-zero-like training: A critical perspective." arXiv preprint arXiv:2503.20783 (2025).

**Questions:**

Please refer to the **Weaknesses** part.

I am looking forward to the authors’ response, and may reconsider this paper based on that.

---

> ### Author Response · Authors · 2025-12-04
>
> Thank you for your thoughtful feedback!
>
> 1. Limited originality and contribution
>
> We agree that other papers have discussed debiasing GRPO by removing standard normalization (e.g. Dr. GRPO, as was cited in the main text) and that some recent large-scale RL efforts have implemented this change, which does limit the novelty of removing standard normalization as a fix for calibration. However, we believe that our work still contributes useful insights to the field.
>
> While Dr. GRPO and GPG both propose debiasing the GRPO policy gradient by removing standard normalization, they do not identify a specific negative impact of the standard normalization term. Dr. GRPO describes a potential question-level difficulty bias due to standard normalization and proposes removing it, but does not report an improvement in performance due to this change (they report equivalent performance with fewer thinking tokens, which appears to be primarily driven by length bias). Similarly, GPG identifies the risk of reward bias due to standard normalization, but explicitly states that they see no performance benefit from removing standard normalization on their accuracy-based task. While removing standard normalization has become increasingly popular, the consequences of this change have remained unclear.
>
> Our work demonstrates that optimization of proper scoring rules with GRPO yields substantially worse average rewards and probability calibration relative to other algorithms. We think this is really important! These results show that in tasks beyond simple accuracy, GRPO can fail catastrophically at optimizing its objective, and that this failure is driven by biased advantage estimates. While the most narrow application of this insight is to say standard normalization should be dropped in GRPO for the probability prediction task, it also highlights an important insight for algorithm design: while unbiased advantage estimates provide consistent optimization signal across tasks, biased advantage estimates need to be well-aligned to the specific objective they are optimizing.
>
> 2. Shallow theoretical analysis
>
> Thank you for your suggestion. We have extended the derivation of the GRPO advantage estimate to no longer assume large group sizes / assume no dependence between prediction and standard deviation, resulting in an additional covariance term in the expression. However, we continue to rely on simulation to build intuition regarding the effect of these policy dependent biases due to the complex interactions present, which we find effective for reasoning about how the standard normalization term causes overconfidence. We agree that a proof of the direction of the effect could be interesting for future work.
>
> 3. Weak experimental design
>
> Thank you for pointing out the lack of clarity regarding the role of the synthetic data experiments. These experiments are intentionally simplified and use idealized models to demonstrate that the GRPO overconfidence phenomenon can be reproduced in a minimal setting and is not dependent on the specifics of the LLM model or architecture.
>
>
> a) We highlight that despite the small group size, the current experiments with full scale LLMs are actually quite compute intensive, requiring ~8xA100 for 12 hours. This limits our ability to run multiple replicates of each large experiment. However, we believe that the strong effect and replication across experiments (2 different models on 2 datasets, synthetic experiments with multiple idealized models and rewards) and theoretical analysis provides strong evidence that the GRPO overconfidence effect is robust.
>
> b) We do agree that it is valuable to highlight what we claim the experiments show (consistent miscalibration and poor rewards when optimizing with GRPO) and what is not demonstrated (claims about small differences in accuracy, that our experiments represent best case performance for each model and can be used to compare overall capacity), and we clarify these points in the text.
>
> 4. Writing and presentation issues
>
> Thanks for the feedback. We have made a number of improvements, including clarifying framing in introduction and conclusion, reordering experiments to better convey the purpose of the synthetic data experiments, moving some experiment details from the appendix to main text, and adding a related works section.
>
> Thanks again for the helpful comments!

---

### Official Review · Reviewer_epXW · 2025-10-31

**Soundness:** 3
**Presentation:** 3
**Contribution:** 2
**Rating:** 4
**Confidence:** 4

**Summary:**

The paper studies 3 types of RL algorithms, PPO, RLOO, and GRPO (with and without group std normalization), for uncertainty estimation in stochastic settings. By evaluating across 3 different scenarios, synthetic, scientific, and medical data, the paper shows that GRPO with group std normalization induces overconfident probability prediction for categorical stochastic outcomes. The paper also provides a theoretical explanation that the std normalization introduces a policy-dependent bias in the advantage estimation, which over-reinforces already confident predictions. A simple solution is to remove the std term.

**Strengths:**

- This paper is well-written and easy to follow. The motivation is well-supported by evidence, including visualizations of the miscalibration and explanations.
- The experimental setup is clear. Although the experiments are not large-scale RL, they cover 3 different datasets and clearly demonstrate the overconfidence phenomenon across different stochastic settings.

**Weaknesses:**

While I appreciate that the authors provide a new perspective on the impact of group std normalization from the lens of uncertainty and overconfidence, the theoretical discussion itself does not bring substantial new insights. Similar ideas, namely that group std normalization can lead to overconfidence, and the corresponding solution of removing the term have also been discussed in [1]. Furthermore, the role of group std normalization has been extensively discussed in prior work, such as Dr. GRPO [2] (discussed in the main text) and other related works [3, 4]. In addition, recent large-scale RL works have also moved away from using group std normalization [5, 6, 7], adopting batch-level normalization or removing the std term entirely. Although the motivations may differ, these existing words nonetheless limit the practical contribution of this paper.

[1] Outcome-based Reinforcement Learning to Predict the Future, arxiv 2025

[2] Understanding R1-Zero-Like Training: A Critical Perspective, COLM 2025

[3] GPG: A Simple and Strong Reinforcement Learning Baseline for Model Reasoning, arxiv 2025

[4] REINFORCE++: An Efficient RLHF Algorithm with Robustness to Both Prompt and Reward Models, arxiv 2025

[5] The Art of Scaling Reinforcement Learning Compute for LLMs, arxiv 2025

[6] Magistral, arxiv 2025

[7] Kimi k1.5: Scaling Reinforcement Learning with LLMs, arxiv 2025

**Questions:**

The main text frequently refers to the appendix for key information. Since there appears to be space left in the main body, it might be better to move some of those into the main text to improve readability.

Overall, while this is a clearly written paper that demonstrates a real and relevant phenomenon, its scope and theoretical discussion are quite narrow, focusing almost entirely on the group std term. Moreover, it does not provide a substantially new or meaningful solution, as simply removing the std term has already been widely adopted in recent RL works. Given the current state of the field, where this topic has already been extensively studied (albeit from different perspectives) and addressed, the paper feels somewhat outdated in scope and timing, and its contribution therefore appears incremental.

---

> ### Author Response · Authors · 2025-12-04
>
> Thank you for your detailed and constructive feedback, especially regarding prior work! We reply to your specific comments and suggestions below. This discussion has been particularly helpful in improving the framing of our paper, and we have added a related works section and updated the framing of our contribution throughout the paper to reflect these discussions.
>
> _Contribution Beyond "Remove Standard Normalization"_
>
> First, we agree that other papers have discussed debiasing GRPO by removing standard normalization (e.g. Dr. GRPO [1], as was cited in the main text) and that some recent large-scale RL efforts have implemented this change, which does limit the novelty of removing standard normalization as a fix for calibration. However, we believe that our work still contributes useful insights to the field.
>
> While Dr. GRPO and GPG [2]  both propose debiasing the GRPO policy gradient, they do not identify a specific negative impact of the standard normalization term. Dr. GRPO describes a potential question-level difficulty bias due to standard normalization and proposes removing it, but does not report an improvement in performance due to this change (they report equivalent performance with fewer thinking tokens, which appears to be primarily driven by length bias). Similarly, GPG identifies the risk of reward bias due to standard normalization, but explicitly states that they see no performance benefit from removing standard normalization on their accuracy-based task. While removing standard normalization has become increasingly popular, the consequences of this change have remained unclear.
>
> Our work demonstrates that optimization of proper scoring rules with GRPO yields substantially worse average rewards and probability calibration relative to other algorithms. We think this is really important! Our results show that in tasks beyond simple accuracy, GRPO can fail catastrophically at optimizing its objective, and that this failure is driven by biased advantage estimates. While the most narrow application of this insight is to say standard normalization should be dropped in GRPO for the probability prediction task, it also highlights an important insight for algorithm design: while unbiased advantage estimates provide consistent optimization signal across tasks, biased advantage estimates need to be well-aligned to the specific objective they are optimizing.
>
> _Relationship to "Outcome-based Reinforcement Learning to Predict the Future"_
>
> Thank you for bringing this paper to our attention! [3] focuses on optimizing reasoning models for a forecasting task and in their main text they do describe observations that GRPO increases overconfidence and standard normalization improves calibration on their task. The effect they observe on calibration by removing standard normalization is relatively small (0.21 to 0.20 ECE, 39% to 13% predictions outside 10%-90% range for GRPO with and without standard normalization).
>
> We believe our work, which was developed independently shortly afterwards (initial submission to a workshop ~1 month afterwards, preprint ~2 months afterwards) provides useful insights beyond the observations in [3]:
>
> 1) We demonstrate that GRPO induces dramatic deterioration of calibration and average rewards across multiple datasets (experiment prediction, medical question answering, and synthetic), reward formulations (log, brier, and spherical scores), and models (Qwen3-4B, Llama3.1-8B-Instruct, and idealized discrete and continuous models). We additionally make the observation that models optimized with GRPO still achieve comparable accuracy based on maximum probability predictions. These results provide clarity on the severity of the bias and help build confidence that it is a general phenomenon.
>
> 2) We provide a theoretical analysis that builds an understanding of why this effect occurs, supporting the conclusions of the empirical analysis and highlighting the challenges of biased advantage estimates.
>
> That said, this paper is certainly relevant to our work, and we have added an explicit discussion of the paper in our related work section.
>
> _Conclusion_
>
> Thank you again for the insightful and helpful comments! Overall, we believe that the insights from our paper go beyond simply re-suggesting to drop group standard normalization to carefully characterizing the consequences of an important failure mode of the still popular GRPO algorithm and highlighting an important design consideration for RL algorithms: while unbiased advantage estimates provide consistent optimization signal across tasks, biased advantage estimates need to be well-aligned to the specific objective they are optimizing.
>
> [1] Understanding R1-Zero-Like Training: A Critical Perspective, COLM 2025
>
> [2] GPG: A Simple and Strong Reinforcement Learning Baseline for Model Reasoning, arxiv 2025
>
> [3] Outcome-based Reinforcement Learning to Predict the Future, arxiv 2025

---

### Official Review · Reviewer_hkZG · 2025-11-07

**Soundness:** 2
**Presentation:** 2
**Contribution:** 2
**Rating:** 4
**Confidence:** 3

**Summary:**

This paper investigates how reinforcement learning methods for training reasoning language models behave when predicting stochastic outcomes, rather than deterministic domains like math. The authors compare GRPO, PPO, and RLOO across synthetic probability prediction tasks, biological perturbation data, and a medical multiple-choice QA dataset. They find that GRPO produces highly overconfident and poorly calibrated probability estimates, whereas other RL algorithms yield well-calibrated results. The paper attributes the miscalibration to GRPO’s group standard normalization step and demonstrates that removing this normalization substantially improves calibration. A theoretical analysis supports that standard normalization introduces a policy-dependent bias that reinforces overconfident predictions.

**Strengths:**

- The paper clearly identifies and isolates the role of group standard normalization in GRPO, providing both empirical and theoretical evidence that this design choice induces systematic overconfidence in stochastic decision settings.

- The experimental evaluation spans synthetic, biological, and clinical knowledge domains, demonstrating that the observed miscalibration behavior persists across qualitatively different tasks and data regimes.

- The theoretical explanation of how standard normalization biases the advantage estimate is well-motivated and aligns with the empirical findings, strengthening the validity and interpretability of the presented results.

**Weaknesses:**

- The empirical evaluation relies only on a single model (Qwen3-4B) across all experiments, which makes it difficult to determine whether the observed calibration differences generalize beyond this specific architecture and scale. Including additional models would substantially strengthen the empirical claims. Furthermore, some details of the experimental setup are under-specified in the main text.

- The model is required to generate explicit natural language tokens to represent  probability, rather than deriving numeric probabilities from token logits. The motivation for this design choice is not sufficiently discussed, and the direct generation approach may conflate reasoning about uncertainty with format adherence. Extracting probabilities from token logits is a common baseline in LM calibration and uncertainty estimation research, including this would provide more comprehensive results.

- The practical benefits of calibrated stochastic outcomes are not fully demonstrated in the presented tasks, since accuracy remains comparable across all methods. While the paper emphasizes calibration as the primary evaluation signal, it is not clearly shown how improved calibration leads to different or better decision-making in downstream applications. Stronger justification or concrete use cases illustrating how calibrated uncertainty materially improves task outcomes would help clarify the broader impact.

**Questions:**

- It would be helpful to elaborate in more detail what is meant by using the log-likelihood of the observed answer under the model’s predicted probability as the reward (Line 151~153). Since this is central to the RL setup, providing a mathematical expression for the reward function in the experiments section would make the methodology clearer for readers.

---

> ### Author Response · Authors · 2025-12-04
>
> Thank you for your helpful feedback! We respond to your suggestions and concerns below
>
> _1a. Empirical evaluation relies on a single model, could strengthen empirical claims with additional models_
>
> Thanks for the suggestion! We extend our experiments to include Llama3.1-8B-Instruct, which represents a different architecture, scale, and post-training pipeline. We observe the same pattern of overconfidence with GRPO across datasets as with the Qwen3-4B model. We additionally clarify the role of the synthetic data experiments, which demonstrate that the same overconfidence phenomenon can be reproduced with minimal non-transformer models and across discrete and continuous probability predictions. These results, along with the theoretical analysis, suggest that the observed overconfidence is a property of the GRPO algorithm and is not dependent on the specifics of the transformer architecture or a specific pretrained model.
>
> _1b. Some experiment details are underspecified in the main text._
>
> We agree! We have moved important experiment descriptions from appendix to main text as well as expanded the description of the MedMCQA experiments.
>
> _2a. Risk of format adherence confounding results when using verbalized uncertainty_
>
> Thanks for the interesting question! We agree that format adherence could confound results. To assess this, we measure the frequency of invalid predictions (misspecified format or invalid probability predictions) in the evaluation set. We find that all models have extremely low rates of invalid predictions: across Qwen and Llama experiments, the biggest frequency was 3 / 4183 invalid predictions, or <0.1%, and that excluding these samples does not meaningfully impact metrics.
>
> _2b. Motivation for verbalized uncertainty_
>
> Thanks for the question! We add a related work section to better contextualize our choice of verbalized uncertainty. Verbalized uncertainty has been a prominent approach to uncertainty estimation with LLMs [1-6]. The benefits of verbalized uncertainty include requiring access to model internals, enabling open-ended response, and not requiring training additional models or repeated sampling. There is also evidence that verbalized uncertainty remains better calibrated after RLHF than token logits [1]. We agree that there is a risk of confounding from format adherence issues, but find that this is rare in our experiments.
>
> _2c. Comparison to analysis of extracting token logits_
>
> Thanks for the suggestion! We update our synthetic experiments to include two idealized models: an Idealized Token Model, which parameterizes probability predictions as a categorical distribution over response tokens (mimicking verbalized uncertainty), and an Idealized Probe Model, which parameterizes probability predictions as a continuous distribution over possible predictions (mimicking continuous probability heads). We demonstrate that the same pattern of overconfidence occurs in both settings. We also note that our theoretical results do not depend on the specifics of how confidence estimates are extracted from the model. Taken together, we believe these results provide strong evidence that the GRPO overconfidence phenomenon arises from the algorithm (specifically the biased advantage estimate), rather than the specific probability extraction mechanism.

---

> ### Author Response · Authors · 2025-12-04
>
> _3. Practical benefits of calibration not clear_
>
> Thanks for this suggestion!
>
> - We update the introduction and related work to more clearly discuss the importance of calibration for downstream optimal decision making [5]
> - We discuss the specific importance of calibrated uncertainty estimates in the two tasks we examine in the experiment section. For example, for medical question answering, there is an important distinction for a medical decision support system between a diagnosis that is 99% confident (may want to treat quickly) and a diagnosis that is 55% confident (may want to invest in further testing), even though the accuracy of the maximum likelihood choice is the same.
> - We additionally discuss why we believe _post-hoc_ calibration is insufficient for addressing miscalibration in the reasoning setting. Our goal is to optimize models to learn to reason about uncertainty itself. If a model interprets weak evidence as decisive, it is unlikely that its reasoning will compose and generalize effectively to new settings, even if _post-hoc_ methods can recover calibration on specific tasks. Additionally, we find that GRPO sometimes collapses probabilities to ties at extreme values and degrades ranking performance, which can limit the effectiveness of these calibration methods.
> - Finally, we also modified the text to better emphasize that models optimized with GRPO achieve dramatically worse average rewards than the other models on this task. Given that the purpose of GRPO is to maximize rewards, it is an issue if the algorithm fails to do so.
>
> 4. Elaborate in more detail on reward functions
>
> Thanks for the suggestion, we have updated the experiments section to explicitly define the precise reward functions.
>
> Thank you again for the thoughtful and constructive feedback!
>
> [1] Just Ask for Calibration: Strategies for Eliciting Calibrated Confidence Scores from Language Models Fine-Tuned with Human Feedback. EMNLP 2023.
> [2] Can LLMs Express Their Uncertainty? An Empirical Evaluation of Confidence Elicitation in LLMs. ICLR 2024.
> [3] Linguistic Calibration of Long-Form Generations. ICML 2024
> [4] Reasoning about Uncertainty: Do Reasoning Models Know When They Don’t Know? arxiv 2025.
> [5] Calibrating Predictions to Decisions: A Novel Approach to Multi-Class Calibration. NeurIPS 2021.

---

### Meta-Review · Area_Chair_MEQg · 2025-12-23

**Summary:**

The reviewers acknowledged the paper's clear identification of how GRPO's group standard normalization induces overconfidence in stochastic settings. However, consensus emerged around critical weaknesses (reflected in scores 4,4,2,2):
* Limited novelty: Removing normalization is well-documented in prior literature and already implemented in recent models
* Narrow experimental scope: Initial experiments used only Qwen3-4B
* Missing contextualization: No related work section positioning findings within calibration research
* Shallow theoretical analysis

While the rebuttal partially addressed concerns (added Llama3.1-8B experiments, clarified design choices), the revisions are insufficient to elevate scores above the acceptance threshold. The core novelty issue remains unresolved. Overall, while I agree that the paper clearly identifies GRPO's overconfidence issue, I also agree with the reviewers that it requires substantial revision before publication: stronger messaging on contribution, comprehensive related work positioning, and concrete demonstration of practical impact beyond applying known techniques.

Note/suggestion to the authors: for binary rewards, RLOO advantages are up to a constant rescaling factor same as those of GRPO without normalization. So effectively, you are claiming/finding that RLOO > GRPO

**Reviewer Concerns:**

The authors added Llama3.1-8B-Instruct experiments showing the overconfidence phenomenon generalizes across architectures. They clarified that simplified experiments isolate the normalization issue as the root cause, not architecture-specific factors. Missing references/discussions were addressed.
Outstanding Concerns:

Practical implications unclear: The authors haven't adequately shown how removing overconfidence affects downstream performance. Need empirical evidence or stronger arguments that the fix improves (or doesn't harm) practical benchmarks.
Limited novelty: While the GRPO failure mode is well-characterized, the fix (removing standard normalization) is a known technique. The contribution needs clearer articulation—either why this specific failure mode is non-obvious/important, or what insights beyond applying existing methods.

Recommendation: Strengthen practical motivation and novelty claims before acceptance. The changes made during rebuttal I believe require one further round of review.

**Reviewer Scores:**

hkZG	4 --> 6:The addition of a second model
epXW	4 -->  4: I think would have been difficult to change their core concern on outdated scope and incremental nature of the solution.
2txt	2  --> 4:
Authors extended the theoretical derivation, but the shallow analysis critique might persist.
GDzm	2 --> 4
The lack of a related work section and the single-model evaluation were addressed. Overall, I think it would have been hard though for the score to go above 4

---

### Decision · Program_Chairs · 2026-01-26

Reject